# High-Power-Density Thermoelectrochemical Cell Based on Ni/NiO Nanostructured Microsphere Electrodes with Alkaline Electrolyte

**DOI:** 10.3390/nano13162290

**Published:** 2023-08-09

**Authors:** Denis Artyukhov, Nikolay Kiselev, Elena Boychenko, Aleksandra Asmolova, Denis Zheleznov, Ivan Artyukhov, Igor Burmistrov, Nikolay Gorshkov

**Affiliations:** 1Department of Power and Electrical Engineering, Yuri Gagarin State Technical University of Saratov, 77 Polytecnicheskaya Street, 410054 Saratov, Russia; ivart54@mail.ru; 2Department of Chemistry and Technology of Materials, Yuri Gagarin State Technical University of Saratov, 77 Polytecnicheskaya Street, 410054 Saratov, Russia; a-asmolova@mail.ru (A.A.); zheleznov_denis@internet.ru (D.Z.); 3Engineering Center, Plekhanov Russian University of Economics, 36 Stremyanny Pereulok, 117997 Moscow, Russia; nikokisely12345@gmail.com (N.K.); elena.boychenko.sar@gmail.com (E.B.); glas100@yandex.ru (I.B.); 4Department of Functional Nanosystems and High Temperature Materials, National University of Science and Technology MISIS, 4 Leninskiy Prospect, 117997 Moscow, Russia; 5N.N. Semenov Federal Research Center for Chemical Physics RAS, Kosygina Street, 4, 119991 Moscow, Russia

**Keywords:** thermoelectrochemical cell, hypothetical Seebeck coefficient, waste heat harvesting, nanostructured nickel hollow microspheres, impedance spectroscopy

## Abstract

Effective low-grade waste heat harvesting and its conversion into electric energy by the means of thermoelectrochemical cells (TECs) are a strong theme in the field of renewable energy investigation. Despite considerable scientific research, TECs have not yet been practically applied due to the high cost of electrode materials and low effectiveness levels. A large hypothetical Seebeck coefficient allow the harvest of the low-grade waste heat and, particularly, to use TECs for collecting human body heat. This paper demonstrates the investigation of estimated hypothetical Seebeck coefficient dependency on KOH electrolyte concentration for TECs with hollow nanostructured Ni/NiO microsphere electrodes. It proposes a thermoelectrochemical cell with power density of 1.72 W·m^−2^ and describes the chemistry of electrodes and near-electrode space. Also, the paper demonstrates a decrease in charge transfer resistance from 3.5 to 0.52 Ω and a decrease in capacitive behavior with increasing electrolyte concentration due to diffusion effects.

## 1. Introduction

Waste heat energy comes from the heat generated by industries such as thermal power plants, steam engines, solar panels and biomass fermentation processes. Typically, their temperature does not exceed 150 °C; therefore, such power is called low-temperature or low-grade due to the limitations of converting such energy into usable energy [1]. Thermoelectrochemical cells (TECs) are the electrochemical analogue of solid-state thermoelectric devices. This type of generator can convert thermal energy into chemical energy, and then to electric energy. Nowadays, it the most promising approach to efficient collection of low-grade heat. The hallmarks of TECs are low maintenance requirements, simple design and no carbon emissions during production and operation [2]. For many years, most researchers have focused on the use of solid-state thermoelectrics, which are based on p- and n-type semiconductors [3]. However, these devices are mostly designed for higher temperatures; therefore, at temperatures below 100 °C their efficiency drops significantly. In this regard, TECs, which are based on electrochemical redox processes and which are more efficient at relatively low temperatures [4], can become an impactful alternative for converting low-grade waste heat [5]. However, at present, thermoelements do not have commercial applications due to the low conversion factor (the most well-known relative Carnot efficiency for a thermoelement is claimed to be 3.95% [6,7]). Nevertheless, owing to recent advances in materials chemistry and electrochemistry of thermoelectric elements, it can be assumed that these devices will be commercially applied in various areas of human life. Thus, in [8], a system is proposed for converting the energy of separated gas flows into electric energy using a vortex tube and a thermoelectrochemical converter. In [9], a wearable system is proposed for collecting the heat energy of the human body (converting the temperature gradient between the body and the environment into electric energy). In [10], a quasi-solid-state TEC for energy-autonomous strain gauge application is described, which can also be used in wearable electronics. The authors of [11] discuss the use of TECs in systems with liquid cooling, particularly, in central processing units or batteries of electric vehicles. In [12], self-powered sensors from four flexible elements connected in series are shown. The sensors can transmit humidity, temperature and air pressure data wirelessly using body heat. In [13], the authors consider several new engineering strategies for expanding the use of thermoelements. Another work [14] is also devoted to the use of thermoelectrochemical cells for power supply of wearable devices from the heat of the human body.

The potential difference in the TEC arises due to the temperature dependence of the electrode potentials and is classical for electrochemical systems described by the isothermal temperature coefficient [15]. For articles published in recent decades, the concept of the “Seebeck coefficient” is widely known. However, due to the fundamentally different mechanisms of the potential difference of semiconductor thermoelectrics and thermoelectrochemical cells, the concept of the “hypothetical Seebeck coefficient” was introduced [16].

In general, the TEC’s hypothetical Seebeck coefficient is of an entropic nature and depends on various key components: ion mobility, electrolyte composition and the loss of charge carriers that occurs at interfaces. The Seebeck coefficient (*S_e_*) can be defined for redox reactions as follows:(1)$$S_{e}=\frac{dV}{dT}=\frac{\Delta S_{B,A}}{nF},$$
where *dV*—potential difference between electrodes, *dT*—temperature difference between electrodes, Δ*S_B,A_*—entropy of a redox pair, *n*—number of electrodes in redox reaction and *F*—Faraday constant.

To enhance the efficiency of TECs, many different structures have been created and widely studied over the past few decades.

The efficiency of TECs largely depends on the nature of the mechanism by which the thermoelectrochemical effect occurs, as well as on the materials used in the cells. The energy generation mechanism can be based on both phase and chemical transformations of electrolytes [17] and redox processes in the electrolyte, while the electrodes are inert [18,19,20]. Another type of thermocells are systems with reversible electrodes [2,21,22]. These systems require a regular change of the thermal gradient direction; however, they show a rather high efficiency level and a hypothetical Seebeck coefficient for systems with aqueous electrolytes [16].

The highest Seebeck coefficient of 9.9 mV·K^−1^ has been demonstrated for a thermoelement containing acetone and isopropanol as the redox pair, where the entropy of acetone vaporization increases the total entropy change in the process of conversion of isopropanol to acetone [17].

Thermoelements based on redox electrolytes are studied most thoroughly, since they can operate continuously throughout their entire service life. For such cells, an aqueous solution of potassium ferro/ferricyanide is used as an electrolyte, making it possible to create a semi-infinite redox system that produces a large entropy of the reaction, which provides a relatively high Seebeck coefficient and a high exchange current [23,24]. In recent years, articles in which the authors are looking for an alternative to the [Fe(CN)_6_]^3−/4−^ system have attracted significant attention. The main reason for looking for a replacement is the potential toxicity of the system, in which, when heated and stirred, HCN gas can be formed [25]. One of the relatively successful examples of replacing the [Fe(CN)_6_]^3−/4−^ electrolyte is the system described in [26]. It should be noted that new systems appear among traditional energy storage devices, which are being developed and can compete with thermoelectrochemical systems [27,28,29,30].

Today, nickel and nickel-based materials are in the most demand in the electrochemical industry due to their outstanding mechanical, chemical and electronic properties, as well as low price compared to Pt electrodes and carbon nanotubes-based electrodes [31,32]. Recently, many advanced electrodes containing nickel-based materials have shown excellent electrochemical performance in fuel cells, batteries and supercapacitors due to fast electron transfer, environmental friendliness, unique surface area, thermal stability, high current density and a better theoretical capacity of about 2584 F·g^−1^ [23,24,25,26,27,28,29,30,31,32,33,34,35]. Owing to their high specific surface area and low cost, metal oxide hollow spheres have received significant attention from large number of researchers in the field [36]. Typically nickel oxide electrodes have been used in strong alkaline electrolytes, such as NaOH or KOH [37]. Nickel nanostructures can be obtained using various methods, such as sol–gel, chemical precipitation, mechanical grinding, electrochemical precipitation and spray pyrolysis [38]. However, spray pyrolysis is the most preferred among the various methods due to it being a simple, cost-effective and time-saving process. In addition, the synthesis does not require reducing and stabilizing chemical agents. Hollow structures increase surface ion diffusion and facilitate electron placement efficiency, resulting in additional pores in the electrode, improving its performance [39]. In our previous work [16], a non-linear dependence of the hypothetical Seebeck coefficient on the temperature difference between the electrodes was established: when the temperature difference reached 30 degrees, the value of the hypothetical Seebeck coefficient of the system drops. This phenomenon can be attributed to entropy or a side reaction of the transition of β-NiOOH to β-Ni(OH)_2_. Yet, almost no attention to the electrochemical processes occurring inside the system at the time of the reaction was paid in that article. In this article, however, we aim to not only investigate the dependence of the hypothetical Seebeck on the temperature difference between the electrodes, but also determine the effect of the electrolyte concentration on both the impedance of the TECs and the energy conversion efficiency.

## 2. Materials and Methods

Synthesis of microspheres was carried out by the means of spray pyrolysis [36,40].

The precursor for the synthesis was an aqueous solution of nickel nitrate (Ni(NO_3_)_2_) with a salt content of 15% (mass). The solution was prepared by adding a weighed amount of crystalline hydrate (Ni(NO_3_)_2_·6H_2_O) to distilled water; water of crystallization was taken into account in the calculations. Next, the solution was filtered through a paper filter to prevent small foreign particles from entering the reactor.

The prepared solution was poured into the container of an ultrasonic generator, which dispersed it from the surface of the solution and created a droplet flow, the constant rate of which (16 L·min^−1^) was maintained by an air compressor. A constant temperature of 900 °C was maintained in the quartz glass reactor (further on, the reactor was replaced with a steel one), which was selected experimentally, taking into account the salt decomposition temperature and the power of the Nabertherm 20/250/13 furnace (Nabertherm GmbH, Lilienthal, Germany) for the highest efficiency of the process.

Next, the reaction mass passed through a filter, which retained solid particles of hollow microspheres. After completion of the process in the reactor and its cooling, the powder was removed from the filter. Since the salt did not decompose completely during pyrolysis, the samples required further calcination in a muffle furnace. The decomposition temperature of nickel nitrate was determined using thermogravimetry. The exposure time was established empirically (in this paper, it was 1 h) and can be varied for scientific purposes to investigate its effect on the dimensional characteristics of the crystallites forming the walls of the microspheres.

Metal microspheres were made from prepared hollow nickel oxide (NiO) microspheres by reduction of nickel oxide with hydrogen in a special setup. Namely, the samples in ceramic crucibles were loaded into a reactor inside a three-section tubular furnace with hydrogen flow. Hydrogen was generated through the process of electrolysis of water in an electrolyzer, which can provide a constant uniform gas flow. The reduction temperature was set at 350 °C, and two-stage gradual heating was used. The degree of completion of the process was controlled by measuring the moisture content of the off-gases.

Powder X-ray diffraction (XRD) was used to determine the crystalline phase. X-ray phase analysis was carried out on an ARL X’TRA diffractometer (Thermo Scientific, Ecublens, Switzerland) using CuKα radiation (λCuKα = 0.15412 nm) in the angle range 2Θ (5–90 degrees). The Rietveld refinement was used to determine the Ni/NiO phase ratio in microsphere samples.

The specific surface area S_sp_ was measured using a NOVA−1200 setup (Quantachromeinstruments, Boynton Beach, FL, USA) using the BET method for low-temperature nitrogen adsorption.

The surface morphology of the synthesized microspheres was studied using scanning electron microscopy (SEM) using an energy dispersive X-ray analyzer (EDX) (Tescan Vega 3, TESCAN, Brno, Czech Republic). The fractional composition of the obtained powders was studied using the laser particle size analyzer Zetasizer Nano ZS (Malvern, United Kingdom) using dynamic light scattering with non-invasive backscattering optics (NIBS).

Hollow nickel microsphere electrodes were made by pressing at 3.5 MPa into pellets with a diameter of 10 mm (2 mm thick) and a mass of 0.5 g. The porous structure of the microspheres was partially preserved due to low pressure, and the approximate volume fraction of pores in the electrode was 96%.

Titanium punches with recesses for sealing O-rings with a diameter of 10 mm were used as current collectors. To increase the temperature difference between the electrodes and provide an increase in output power, porous sponges and membranes are often used as a thermal separator. In this study, non-woven geotextile fabric TU 8397-056-05283280−2002 was chosen as a thermal separator.

The electrodes were placed in the body of the TEC and pressed to the current collectors by a thermal separator. A schematic illustration of a TEC is shown in Figure 1.

The temperature difference in the cell was created using Peltier elements of the type TEC1−12706. The Peltier elements were powered by a laboratory power supply Element 3010D (Element, Moscow, Russia). The body of the TEC was carved from polytetrafluoroethylene−4. The final distance between the electrodes was 10 mm.

Aqueous electrolytes were prepared by dissolving KOH (99.5% purity, Vekton Ltd., St. Petersburg, Russia) in distilled water. The actual temperature on both sides of the cell was measured using thermocouples (OMEGA type-T), which were connected through a Fluke 8846A digital multimeter (Fluke Corporation, Everett, WA, USA). The current–voltage curves of the cell were measured in the potentiodynamic mode at a scanning rate of 1 mV·s^−1^ using a PI-50-PRO potentiostat (Elins, Chernogolovka, Russia). The initial voltage was set to open circuit, which was measured in the voltmeter mode, and the complementary value was set to 0 V. The measurements were carried out in the temperature range from 10 to 50 K in 10 degree steps at electrolyte concentrations of 1 mol·L^−1^, 2 mol·L^−1^, 4 mol·L^−1^ and 6 mol·L^−1^. The temperature of the cold electrode was 293 K. The impedance of the thermoelectrochemical system was measured with an Elins Z-500Pro impedance meter (Elins, Chernogolovka, Russia) in the frequency range from 0.1 Hz to 50 kHz with an amplitude of 5 mV.

## 3. Results and Discussion

Figure 2 shows the XRD pattern of synthesized Ni/NiO microspheres. All reflections fully correspond to Ni and NiO phases, space group Fm–3m.

The Rietveld method was used to refine the parameters of the crystal lattice of the synthesized microspheres and to determine the Ni/NiO phase ratio. Figure 2 shows the difference (blue line) between the experimental (black balls) and calculated (red line) diffraction patterns. A slight discrepancy is observed for the reflections with the highest intensity. The structural parameters of the unit cells of the Ni phases were determined: a = b = c = 3.5212 Å, α = β = γ = 90°, unit cell volume 43.658 Å^3^ and theoretical density 8.931 g·cm^−3^, and for NiO a = b = c = 4.177 Å, α = β = γ = 90°, unit cell volume 72.881 Å^3^ and theoretical density 6.808 g/cm^3^. The rather low chi-factor χ^2^ = 1.757 indicates a high level of agreement between theoretical and experimental data. The Ni/NiO phase ratio in the microspheres is estimated to be 78.1/21.9.

The images (Figure 3a,b) made using scanning electron microscopy method demonstrate the structure and size of the synthesized microspheres. The average size of Ni/NiO spheres is about 0.4 μm, with a significant proportion of unshaped particles, as well as chipped and deformed spheres. The presence of cavities in the microspheres is proved by the images of chipped microspheres, in which the walls and the cavity inside are clearly visible. A more detailed image (Figure 3c) shows that the walls of hollow microspheres are composed of individual nanoparticles with diameters of about 50 nm, which makes the synthesized materials not just microspheres, but nanostructured ones. Particle size analysis (Figure 3c) confirms the significant presence of nanosized particles.

The specific surface area was measured using the low-temperature nitrogen adsorption method. The value of the specific surface was 23.191 m^2^·g^−1^, which is an average value for this type of sample in comparison with those described in the literature [41].

The presence of hollow structures in microspheres significantly increases the surface area of the electrode, which leads to an acceleration of ion exchange reactions and an increase in current density. At a relatively low electrolyte concentration, the short-circuit current density is about 700 µA·cm^−2^ (Figure 4a) compared to 1 µA·cm^−2^ obtained for flat electrodes cut from metallic nickel foil in [21]. As shown in Figure 4, the open-circuit potential increases with larger temperature difference; however, the shape of the I–V curves differs from the typical linear shape. Meanwhile, the short-circuit current density at low electrolyte concentrations has a small temperature dependence, which clearly intensifies with increasing electrolyte concentration (Figure 4c,d).

Figure 5 shows the graphs of the specific power generated by thermoelectrochemical cells in terms of energy per 1 square meter of electrode surface area. It can be noted that with a rise in temperature and electrolyte concentration, the specific power grows—at the minimum point (ΔT = 10 K, 1 mol·L^−1^) the specific power is 400 mW·m^−2^, and at the maximum point (ΔT = 50 K, 6 mol·L^−1^) it is 1720 mW·m^−2^.

It should be pointed out that the shape of the open-circuit voltage on the temperature difference dependence curve (Figure 6) is not typical for thermoelectric systems, since the Seebeck coefficients of the systems behave non-linearly. Based on this, one can assume that the interfacial transition described in [16] plays a significant role in the non-linearity of the Seebeck coefficient of these systems.

A peculiar phenomenon is that although the open-circuit voltage of the cell naturally increases with rising temperature difference between the electrodes, with varying electrolyte concentration, a clear growth in voltage is observed at a concentration of 2 mol·L^−1^. However, despite the fact that the system with an electrolyte concentration of 2 mol·L^−1^ has the highest voltage values, the highest power density is observed at an electrolyte concentration of 6 mol·L^−1^, which can be explained by a significant rise in the current density of the system.

For a deeper understanding of the processes occurring inside the thermoelectrochemical cell, a study was carried out by the means of impedance spectroscopy. Figure 7 shows the Nyquist plot of TECs with Ni/NiO electrodes at different temperatures with varying electrolyte concentrations as well as equivalent circuits.

Electrodes based on nickel spheres are similar to the cathode of a Ni-Cd element, therefore, the electrochemical reactions on the hot and cold electrodes of the thermoelectrochemical cell are identical to the charging and discharging for the corresponding electrode. So, for a hot electrode, a corresponding reaction is similar to the charging one:β-Ni^2+^(OH)_2_ + OH^−^→ β-Ni^3+^O^2−^OH^−^ + H_2_O + e^−^(2)

In its turn, the cold electrode reaction is equivalent to the discharging reaction:β-Ni^3+^O^2−^OH^−^ + H_2_O + e^−^ → β-Ni^2+^(OH)_2_ + OH^−^(3)

Thus, the electrochemical redox reaction of the cold electrode can be described as follows: Ni^3+^ is reduced to Ni^2+^, while hydroxide ions (OH^−^) are transferred to the hot electrode, where oxidation from Ni^2+^ to Ni^3+^ occurs. According to the processes occurring at the electrode–electrolyte interface as well as inside the electrode and electrolyte, the impedance of the TEC can be interpreted by an equivalent circuit (Figure 7e). The first series resistance element R1 corresponds to the electrolyte resistance, the section with the parallel connection of the constant phase element CPE1 and the resistance R2 corresponds to the accumulation of the electric double layer capacitance and the charge transfer resistance reflecting the exchange between ions and electrons at the interfaces of the electrodes. The last element of the constant phase CPE2 corresponds to the electrochemical capacitance accumulated due to redox reactions [42].

Table 1 shows the equivalent circuit parameters for the system.

Table 1 shows that an increase in electrolyte concentration leads to a clear decrease in its resistance R1. Figure 8 shows graphs of charge transfer resistance R2 dependencies on electrolyte concentration at various temperature gradients. It can be noted that a higher electrolyte concentration leads to a decrease in the resistance R2. However, with a growing temperature gradient, a small increase in resistance is observed at each concentration, which can be explained by the rising entropy of the system. Yet, the simultaneous decline in contact resistance R1 and charge transfer resistance R2 leads to an increase in system currents, which is shown in Figure 4.

The constant phase element (CPE) allows one to describe the nature of the conductivity in the system. So, if n = 0, the element represents an ideal resistance; if n = 1, it is an ideal capacitor. The value n = 0.5 characterizes diffusion processes (Warburg impedance). Although in the classical representation the CPE1 element should have a capacitive nature, from the results demonstrated in Table 1, it can be seen that it is of a clear diffusion character.

The value of n1 remains within ±0.5, which means that diffusion character remains unchanged. Although, with a more detailed analysis of the n2 component, it becomes clear that with an increase in the electrolyte concentration, the limiting capacitance gradually acquires a diffusion character (Figure 9), ceasing to be close to ideal. P2 values correspond to ~0.1 F, which distinctly indicates the occurrence of an electrochemical reaction [43].

It is widely known that the Seebeck coefficient of the system has a pronounced entropic nature [17,44,45]. With an increase in the alkali concentration, the interaction of the oxide phase of the Ni/NiO spheres with OH^−^ ions should also be enhanced. In thermoelectrochemical cells, this is demonstrated by a growing current density and power with an increase in the KOH concentration in the electrolyte. In this case, the change in potentials, as well as the regularity of the intensity of the interaction of nickel spheres on cold and hot electrodes, according to the equations presented above, has a more complex dependence on the concentration. This explains the absence of a direct dependence in the values of the Seebeck coefficient and, accordingly, in the values of the open-circuit voltage at various temperature gradients between the electrodes and at various electrolyte concentrations.

## 4. Conclusions

In this work, it has been shown that the hypothetical Seebeck coefficient in thermoelectrochemical cells with Ni/NiO electrodes behaves non-linearly, both with a change in temperature between the electrodes and with a change in the concentration of KOH electrolyte. Due to the improved body design, the developed TECs can provide open-circuit voltages up to 258 mV at a temperature difference between the electrodes of ΔT = 50 K. In addition, it was made possible to increase the short-circuit current density to 1.07 mA·cm^−2^, which is an extremely high value for liquid systems. Taking into account high values of currents and voltages, the maximum power density was as high as 1.72 W·m^−2^. However, the highest values of the Seebeck coefficient in the system were observed at small temperature gradients, which makes the described system the most practical for powering wearable gadgets using human body heat.

## Figures and Tables

**Figure 1 nanomaterials-13-02290-f001:**
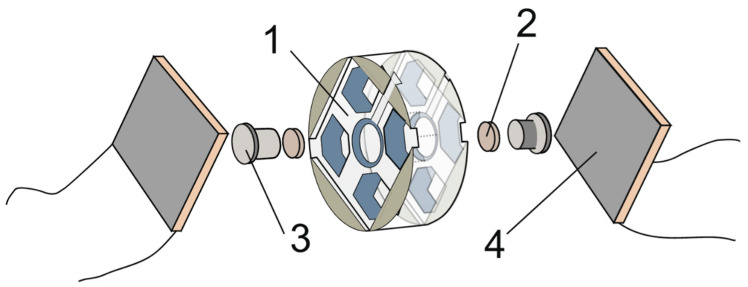
Images of TEC components. 1—PTFE body; 2—Ni/NiO electrode; 3—collecting electrode; 4—Peltier element TEC1−12706.

**Figure 2 nanomaterials-13-02290-f002:**
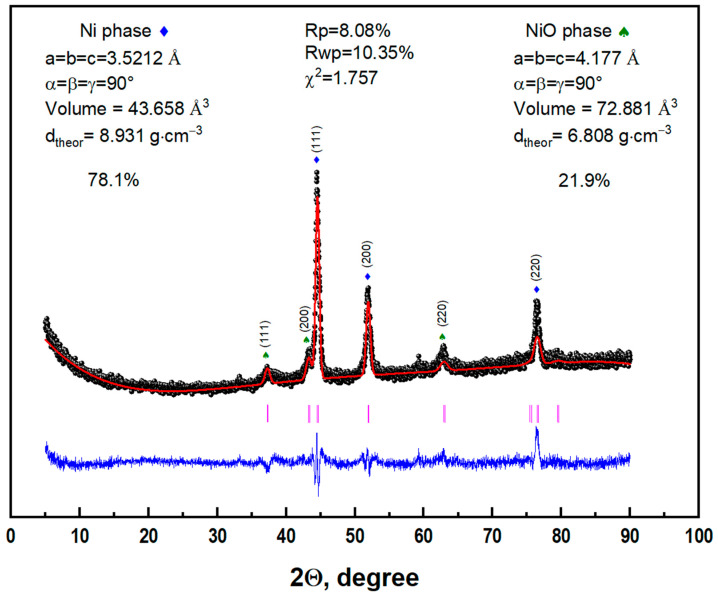
Refined Rietveld plot showing experimental data (black balls), calculated data (red line) and difference (blue line) for Ni/NiO microsphere powder.

**Figure 3 nanomaterials-13-02290-f003:**
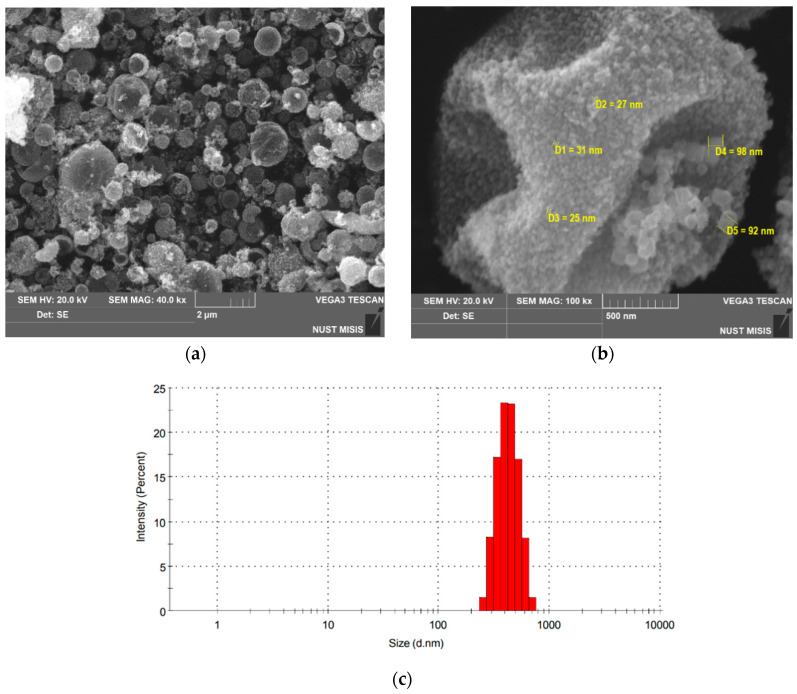
(**a**) SEM image of synthesized hollow Ni/NiO microspheres, (**b**) nanostructured Ni/NiO microsphere, (**c**) particle size distribution of Ni/NiO microspheres.

**Figure 4 nanomaterials-13-02290-f004:**
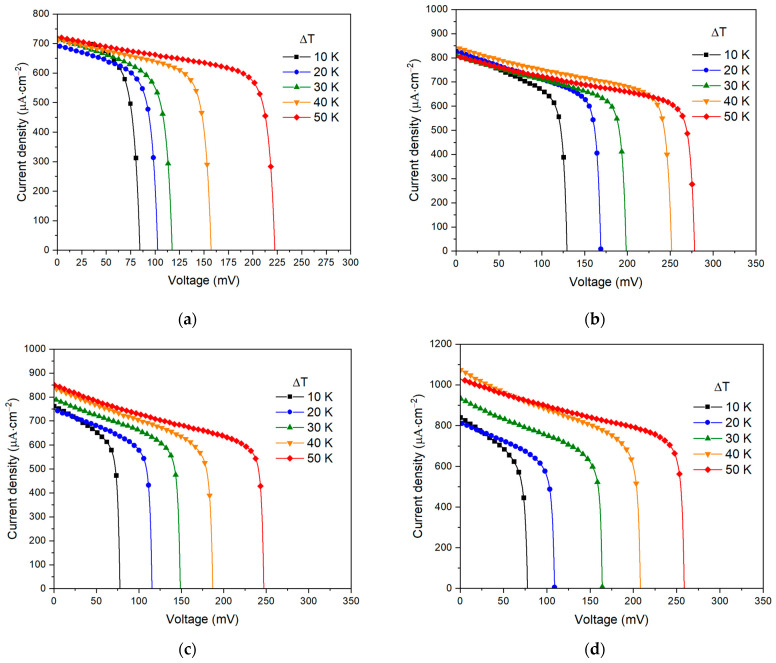
Short-circuit current density dependence on open-circuit voltage for electrolyte concentration: (**a**) 1 mol·L^−1^, (**b**) 2 mol·L^−1^, (**c**) 4 mol·L^−1^, (**d**) 6 mol·L^−1^.

**Figure 5 nanomaterials-13-02290-f005:**
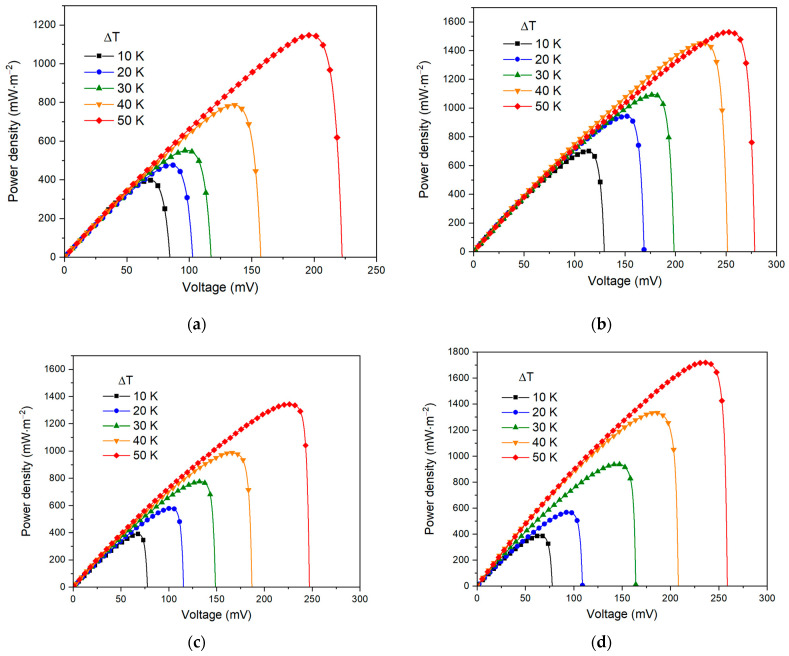
Output power curves on cell voltage for electrolyte concentration: (**a**) 1 mol·L^−1^, (**b**) 2 mol·L^−1^, (**c**) 4 mol·L^−1^, (**d**) 6 mol·L^−1^.

**Figure 6 nanomaterials-13-02290-f006:**
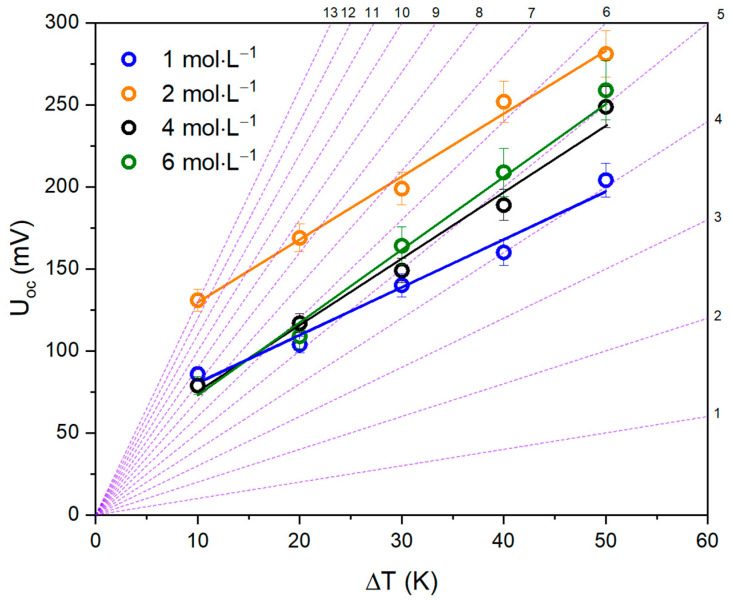
Open-circuit voltage dependence on temperature difference (violet lines represent the ideal Seebeck coefficient graphs).

**Figure 7 nanomaterials-13-02290-f007:**
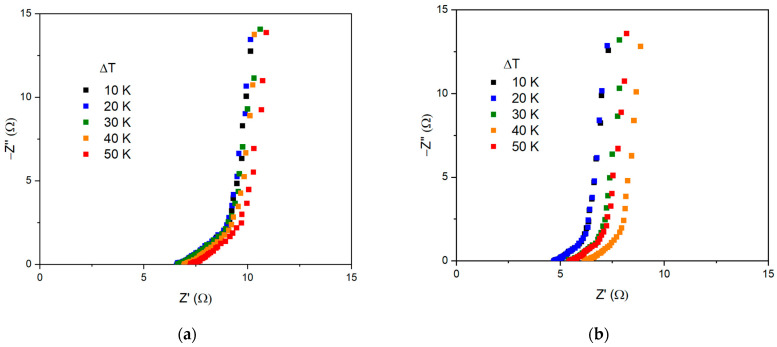
Total impedance of the cell: (**a**) 1 mol·L^−1^, (**b**) 2 mol·L^−1^, (**c**) 4 mol·L^−1^, (**d**) 6 mol·L^−1^, (**e**) equivalent circuit of TECs.

**Figure 8 nanomaterials-13-02290-f008:**
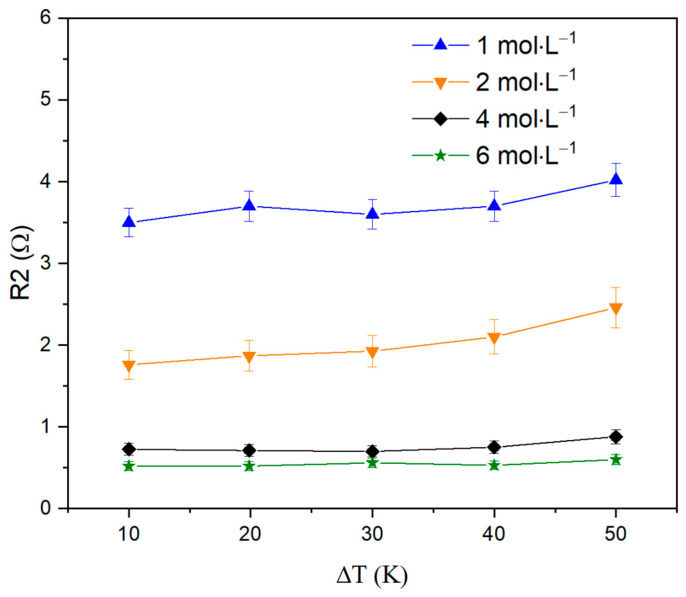
Charge transfer resistance R2 versus electrolyte concentration graphs at various temperature gradients.

**Figure 9 nanomaterials-13-02290-f009:**
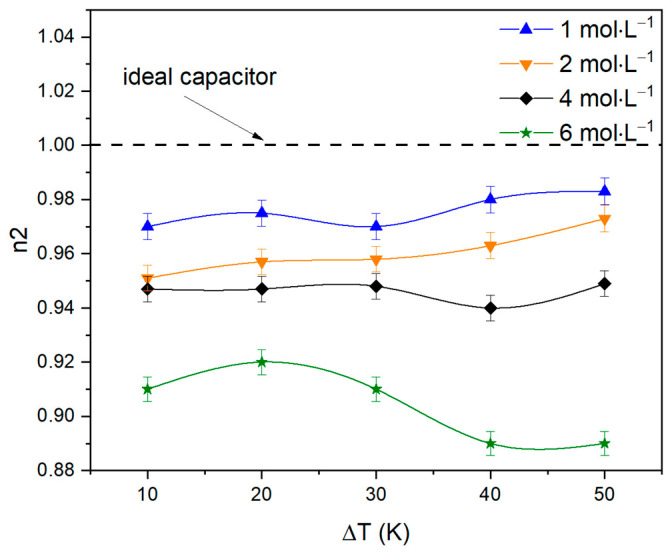
n2 component versus electrolyte concentration graphs at various temperature gradients.

**Table 1 nanomaterials-13-02290-t001:** Parameters of equivalent circuits for TECs.

ΔT, °C	R1, Ω	R2, Ω	P1	n1	P2	n2
1 mol·L^−1^						
10	6.7	3.5	0.09	0.47	0.124	0.97
20	6.6	3.7	0.09	0.47	0.115	0.975
30	6.63	3.6	0.09	0.45	0.11	0.970
40	7	3.7	0.09	0.49	0.115	0.98
50	7.2	4.02	0.09	0.488	0.112	0.983
2 mol·L^−1^						
10	4.7	1.76	0.09	0.52	0.12	0.951
20	4.69	1.87	0.09	0.496	0.118	0.957
30	5.43	1.926	0.09	0.5	0.116	0.958
40	6.22	2.1	0.09	0.5	0.117	0.963
50	5.476	2.458	0.09	0.528	0.114	0.973
4 mol·L^−1^						
10	3.1	0.725	0.12	0.366	0.11	0.947
20	3.5	0.71	0.12	0.41	0.11	0.947
30	3.7	0.7	0.12	0.474	0.108	0.948
40	3.58	0.75	0.12	0.47	0.1	0.941
50	3.08	0.88	0.12	0.41	0.1	0.949
6 mol·L^−1^						
10	3.4	0.52	0.09	0.46	0.11	0.91
20	3.22	0.52	0.09	0.5	0.1	0.92
30	2.05	0.56	0.09	0.5	0.11	0.91
40	2.01	0.53	0.09	0.47	0.12	0.89
50	2.36	0.6	0.09	0.49	0.12	0.89

## Data Availability

The data presented in this study are available in this article.

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
