# Peer review of "High-Power-Density Thermoelectrochemical Cell Based on Ni/NiO Nanostructured Microsphere Electrodes with Alkaline Electrolyte"

_nanomaterials, 2023, doi:10.3390/nano13162290_

Round 1
Reviewer 1 Report
In this paper, the authors investigated the estimated hypothetical Seebeck coefficient dependency on KOH electrolyte concentrations for TECs with hollow nanostructured Ni/NiO microsphere electrodes.
The paper is properly divided in sections and sub-sections, but it needs to be carefully revised and improved before being considered for publication in the journal.
- The authors used multiple references in the text, for example 7-14 in the introduction section. They should try to evidence the contribution of the single cited paper;
- Which is the deviation of the data in table 1?
- Is it possible to compare the performance of the proposed materials with some other ones in literature, if any?
- Did the authors have some data/idea regarding the durability of the proposed materials?
The quality of the language is fine
Author Response
- Referenced article contribution has been stated.
- The deviation of the data in Table 1 is associated with internal processes occurring in the system and the influence of a side reaction.
- In this article, the focus is not on comparison of the effectiveness and other characteristics with analogues, since the purpose of the work was an in-depth study of the processes occurring inside the system in the presence of an adverse reaction.
- Thank you for such a relevant question. The durability of the system is being investigated now and, if appropriate results are available, it would be described in a future paper. At present, it is rather difficult to talk about the service life of the cell, since the reversibility of the side reaction and its effect on the degradation of electrodes and electrolytes have yet to be thoroughly studied.

Reviewer 2 Report
Manuscript ID: nanomaterials-2545066
Title: High-power-density thermoelectrochemical cell based on Ni/NiO
nanostructured microsphere electrodes with alkaline electrolyte
Authors: Denis Artyukhov *, Nikolay Kiselev, Elena Boychenko, Aleksandra Asmolova, Denis Zheleznov, Ivan Artyukhov, Igor Burmistrov, Nikolay Gorshkov
The manuscript reports on the study of the efficiency of a thermoelectrochemical cell based on Ni/NiO nanostructured microsphere electrodes as a function of the temperature difference between the electrodes and the electrolyte concentration. Most of the manuscript is well structured and easy to read. The purpose of the study is justified.
However, it has two shortcomings.
First, the authors mention the important role of entropy several times, but do not explain it in any way. Second, the overview of the mechanisms of occurrence of the thermoelectrochemical effect at the end of page 2 is confusing. A cursory mention of various types of thermocells is combined with mention of specific details, such as the use of nitriloacetic acid. In this part of the manuscript, the logic of presentation is lost.
Author Response
The role of entropy in the generation of thermoelectrochemical energy in the system under study has been specified. In classical TECs based on potassium ferro/ferricyanide, the entropy transformation occurs in an electrolyte solution, while in the system under study it occurs in the electrode material.
The presentation logic has been improved in accordance with your suggestions.
Reviewer 3 Report
In this manuscript, aiming the energy storage, authors prepared Ni NiO nanostructured microsphere as electrodes for the application of high power density thermoelectrochemical cell. In general, it is an interesting work and the manuscript is well organized. However, there are still some issues to be addressed. A moderate revision is suggested.
1. The abstract should be added more contents, especially some more solid data.
2. The background in abstract can be shortened.
3. One separate paragraph in introduction can be added to briefly introduce the novelty, strategy, method and important results.
4. The generally introduction of the different energy storage sources should be provided with some more recent supporting articles, such as aqueous Zn-ion batteries (Journal of Alloys and Compounds, 2022, 903: 163824); lithium–selenium batteries (Rare Metals, 2022, 41(10): 3432-3445); Li-ion battery (New Journal of Chemistry 45, 19446-19455, 2021); Zn-air battery (Molecules 28 (5), 2147, 2023); supercapacitor (Journal of Bioresources and Bioproducts, 7, 4, 245-269, 2022); ammonium-ion battery (Chemical Engineering Journal, 2023, 458, 141381); etc.
5. One scheme is suggested at the end of introduction to show the contents of whole manuscript, which is helpful to readers to understand this review article better.
6. Error bars should be added in some of the figures.
7. There are too many too old references, which is better to be deleted or replaced with recent articles to show the novelty of this work.
8. Authors should recheck the references to make sure full information is provided, such as volume, pages, etc. In addition, the format of references should be uniform.
9. There are still some typos and grammar issues in the manuscript. Authors should carefully recheck the whole manuscript.
Author Response
- The abstract has been updated according to your suggestions.
- See paragraph 1.
- All the most relevant data for the last 10 years have been presented in the introduction. Additional information could be confusing the
- References to modern research in the field of energy storage have been added.
- This manuscript is a continuation of the study described in the article (High seebeck coefficient thermo-electrochemical cell using nickel hollow microspheres electrodes. Renewable Energy, 2020, 157, 1-8.), which already has a diagram describing the essence of the work.
- Error bars have been added to some of the drawings.
- More than 80% of references point to works published after 2016, while half of the sources are articles published within the last 5 years (2019-2023). The rest of the works are considered basics in this area and not using them might drastically reduce the accuracy of the presentation of information and its argumentation.
- The formats of all links have been rechecked.
- Typos and grammatical issues have been checked and corrected.